



# A hydroclimatic model for the distribution of fire on Earth

Matthias M. Boer[1], Víctor Resco De Dios[2], Elisa Z. Stefaniak[1], Ross A. Bradstock[3]

[1] Hawkesbury Institute for the Environment, Western Sydney University, Richmond, Australia.
[2] Department of Crop and Forest Sciences-AGROTECNIO Center, Universitat de Lleida, E 25198 Lleida, Spain.
[3] Centre for Environmental Risk Management of Bushfires, University of Wollongong, Wollongong, Australia.

*Correspondence to*: Matthias M. Boer (m.boer@westernsydney.edu.au)

**Abstract.** The distribution of fire on Earth has been monitored from space for several decades, yet the geography of global fire regimes has proven difficult to reproduce from interactions of climate, vegetation, terrain and land use by empirical and process-based fire models. Here, we propose a simple, yet robust, model for global fire potential based on fundamental
biophysical constraints controlling fire activity in all biomes. In our 'top-down' approach we ignored the dynamics of individual fires and focus on capturing hydroclimatic constraints on the production and (seasonal) desiccation of fuels to predict the potential mean annual fractional burned area, here estimated by the 0.99 percentile of the observed mean annual fractional burned area ($F_{0.99}$). We show that 80% of the global variation in $F_{0.99}$ can be explained from a combination of mean annual precipitation and potential evapotranspiration. The proposed hydroclimatic model reproduced observed fire activity
levels equally well across all biomes and provided the first objective underpinning for the dichotomy of global fire regimes in two domains characterised by either fuel production limitations on fire or fuel dryness limitations on fire. A sharp transition between the two climate-fire domains was found to occur at a mean annual aridity index of 1.9 (1.94±0.02). Our model provides a simple but comprehensive basis for predicting fire potential under current and future climates, as well as an overarching framework for estimating effects of human activity via ignition regimes and manipulation of vegetation.

## 1 Introduction

Satellite-based Earth observation is providing an increasingly accurate picture of global fire patterns (Roy et al., 2005;Giglio et al., 2013;Robinson, 1991;Chuvieco and Martin, 1994). The highest fire activity is observed in seasonally dry (sub-)tropical environments of South America, Africa and Australia, but fires occur with varying frequency, intensity and seasonality in
almost all biomes on Earth (Archibald et al., 2013). The particular combinations of these fire characteristics, or fire regimes (Gill, 1975), are known to emerge from the combined influences of climate, vegetation, terrain and land use, but their global distribution has so far proven difficult to reproduce by mathematical models (Hantson et al., 2016). Most current land surface models (LSMs) and Dynamic Global Vegetation Models (DGVMs) have some capacity to simulate fire activity from basic environmental variables but predictions (Hantson et al., 2016) usually only agree with observations in some biomes (e.g.
savannas of the Sub-Saharan Africa), while disagreeing in others (e.g. boreal forests of North-America). Humans add further



complexity to global fire patterns by, amongst others, changing vegetation community composition, structure and flammability, active use of fire to clear land or reduce fire hazard, and fire suppression (Knorr et al., 2016;Archibald et al., 2012;Parisien et al., 2016;Bowman et al., 2011). The partial success of current models in predicting global fire patterns suggests that their fire modules fail to capture some aspects of the biophysics that control fire activity across different environments.

Incomplete understanding of biophysical drivers and constraints that underlie current global fire patterns creates uncertainty in model predictions of how fire regimes, fire-prone ecosystems and related biogeochemical cycles may respond to rising atmospheric $[CO_2]$ and climate change (Harris et al., 2016).

Here, we go back to the fundamental biophysics of fire to formulate a simple, yet robust, model for the prediction of potential
mean annual levels of fire activity across all global biomes. For this study we define fire activity in terms of the mean annual fractional burned area, $F$, of the landscape or land area unit, ranging from 0 in long unburnt landscapes to 1 in landscapes that burn completely on an annual basis (Giglio et al., 2013). By predicting $F$ we ignore the dynamics of individual fires, but this is appropriate in our view given the fact that most fires are orders of magnitude smaller (e.g., Archibald et al., 2013;Malamud et al., 2005) than the typical grid size (e.g. 0.5° x 0.5°) of LSMs and DGVMs.


Building on Bradstock's (2010) 'four-switch' concept we assume that four fundamental conditions need to be met for a landscape fire to occur: i) there must be enough plant biomass (i.e. fuel) to carry a fire, ii) the extant fuel must be dry enough to be ignitable, iii) weather conditions need to be favourable (i.e. hot, dry and windy) for a fire to spread, and iv) there must be an ignition. Bradstock (2010) conceptualized these conditions as four 'switches' in a series circuit that need to be 'on' for
a fire to occur. While any fire will require alignment of all four switches at some point in time, in the context of modelling global fire patterns we emphasize that the four switches operate at disparate rates, with an associated hierarchy of conditional constraints on fire: production of plant biomass and build-up of fuel loads occurs over months to years, fuels dry out over weeks to months, and fire weather varies over time scales of hours to days, while ignitions are instantaneous events. Therefore, we hypothesize that the mean annual fractional burned area ($F$) can be predicted from long-term fuel production and fuel
drying rates (i.e. switches 1-2), while information on fire weather and ignitions (i.e. switches 3-4) is only required when modelling the specific attributes of individual fires (e.g. size and burn pattern). The hierarchical organization of the four switches further implies that the fractional burned area predicted from long-term fuel production and fuel drying rates alone represents an upper limit of fire activity that is only reached when fire weather and ignition limitations are minimal.

To test the hypothesis that the upper quantiles of $F$ can be predicted from long-term fuel production and drying rates, we analysed global burned area data together with indices of fuel productivity and fuel dryness, both calculated from the climatic water balance (Stephenson, 1998). Building on the methods developed by a previous study of Australian fire regimes (Boer et al., 2016), we propose a new global model that predicts the upper limit (i.e. the 0.99 quantile) of the mean annual fractional burned area, $F_{0.99}$, from two basic hydroclimatic variables: mean annual precipitation, $P$, and potential evapotranspiration, $E_0$.





The model also predicts the relative importance of either fuel productivity or fuel dryness constraints on $F_{0.99}$, which we hypothesized to vary with the global distribution of land cover types and corresponding fuel types. Consistent with the variable constraints hypothesis (Krawchuk and Moritz, 2011), in dryland environments with grassy fuels we expected $F_{0.99}$ to be primarily limited by fuel productivity constraints, while in more mesic environments dominated by woody vegetation and litter fuels $F_{0.99}$ was expected to be limited primarily by fuel dryness constraints. Finally, we explored whether the global

distribution of contemporary fire regime classes or 'pyromes' as classified by Archibald et al. (2013) on the basis of four fire regime metrics (i.e. fire return interval, maximum fire intensity, length of fire season, maximum fire area.) was associated with the relative importance of fuel productivity or fuel dryness constraints on $F_{0.99}$.

## 2 Materials and methods

**2.1 Modelling approach**

In this study our aim was to predict global patterns of potential mean annual fractional burned area ($F_{0.99}$) as a function of basic biophysical constraints on the production and dryness of fuel material. Following Boer et al. (2016), we assumed that both the production and drying of fuel material are essentially functions of the local water and energy budgets available for the production and desiccation of plant biomass. The long-term climatic water balance, calculated from mean annual $P$ and

$E_0$, captures these interactions of biologically available water and energy (Stephenson, 1998). When calculated over long time scales (>>years) and broad spatial scales (>>km$^2$), changes in the soil water store and lateral water inputs can be assumed to be negligible so that the climatic water balance is reduced to: $P - Q - E = 0$, where $Q$ is runoff/drainage losses and $E$ is actual evapotranspiration. $E$ is a reliable predictor of continental patterns of annual primary productivity at annual timescales (Rosenzweig, 1968;Yang et al., 2013) and therefore a reasonable proxy for fuel production rates (Meentemeyer et al.,

1982;Matthews, 1997). Similarly, the potential for drying of fuel material can be assumed to be proportional to the atmospheric moisture demand ($\propto E_0$) that cannot be met by available water ($\propto E$), which is the climatic water deficit as defined by Stephenson (1998), $D = E_0 - E$. For long timescales and large land areas, $E$ can be estimated from $P$ and $E_0$ using the semi-empirical Budyko curve (Budyko, 1958;Zhang et al., 2010;Williams et al., 2012):

$$E = P \sqrt{\frac{E_0}{P} \tanh\left(\frac{E_0}{P}\right)^{-1} \left[1 - \exp\left(-\frac{E_0}{P}\right)\right]} \tag{1}$$


Recently, Boer et al. (2016) demonstrated that continental patterns of $F_{0.99}$ in Australia can be accurately predicted from mean annual $E$ and $D$. Since most global fire regime classes are represented in Australia (Murphy et al., 2013;Archibald et al., 2013), we hypothesized that global $F_{0.99}$ can also be modelled as a function of mean annual $E$ and $D$. Here we present a new global



$F_{0.99}$ model that consists of two flexible sigmoidal functions, $F_{0.99}(E)$ and $F_{0.99}(D)$, describing the increase in $F_{0.99}$ with mean

annual actual evapotranspiration ($E$) and climatic water deficit ($D$), respectively:

$$F_{0.99}(E) = \left[1 + \frac{E_2 - E}{E_2 - E_1}\right]\left[\frac{E}{E_2}\right]^{\frac{E_2}{(E_2 - E_1)}} \quad \text{with } 0 < E \leq E_2$$

$$F_{0.99}(E) = 1 \quad \text{for } E > E_2 \tag{2}$$

$$F_{0.99}(D) = \left[1 + \frac{D_2 - D}{D_2 - D_1}\right]\left[\frac{D}{D_2}\right]^{\frac{D_2}{(D_2 - D_1)}} \quad \text{with } 0 < D \leq D_2$$

$$F_{0.99}(D) = 1 \quad \text{for } D > D_2 \tag{3}$$

$$F_{0.99}(E, D) = F_{max} F_{0.99}(E) F_{0.99}(D) \tag{4}$$

where $F_{max}$ is the global maximum of $F_{0.99}$, here set at 1. The shape of $F_{0.99}(E)$ and $F_{0.99}(D)$ is set by two parameters (Yin et al., 2003): $E_1$ (or $D_1$) is the value at which $F_{0.99}$ increases most strongly with $E$ (or $D$), and $E_2$ (or $D_2$) the value at which $F_{0.99}$ becomes irresponsive to further increase of $E$ (or $D$). Since the two predictor variables, $E$ and $D$, are not independent, the fitted

$E, D, F_{0.99}$ response surface was mapped to (orthogonal) axes of $P$ and $E_0$ before interpreting the shape of the response surface in terms of biophysical constrains on global fire activity.

### 2.2 Data

#### 2.2.1 Burned area

Global annual burned area data at 0.25° x 0.25° spatial resolution for the period July 1995 – June 2016 were obtained from the GFED4 database (Giglio et al., 2013). The mean annual fractional burned area, $F$, was calculated by summing the burned areas within each grid cell for the entire observation period and dividing by the area of the grid cell and the duration of the observation period.

**2.2.2 Climatic water balance**

Gridded mean annual precipitation, $P$, was obtained from WorldClim (Hijmans et al., 2005), while gridded mean annual potential evapotranspiration, $E_0$, based on the Hargreaves method (Zomer et al., 2007;Zomer et al., 2008), was obtained from the Global Aridity and Potential Evapotranspiration Data base at CGIAR-CSI ([http://www.cgiar-csi.org/data/global-aridity-](http://www.cgiar-csi.org/data/global-aridity-)



and-pet-database). Both data layers are based on observations over the 1950-2000 period and have a spatial resolution of 30

arcseconds; both were resampled to the 0.25° x 0.25° grid of the GFED4 data base using bilinear interpolation. Mean annual

actual evapotranspiration, $E$, was predicted from $P$ and $E_0$ using the Budyko curve (Eq. 1).

### 2.2.3 Landcover and fuel types

This study focused on areas of (semi-)natural vegetation. The corresponding vegetation mask was constructed from the Land

Cover Type product (MCD12Q1) of the Moderate Resolution Imaging Spectroradiometer (MODIS) by first resampling the

data layer to the 0.25° grid of the GFED4 data base, using nearest neighbour interpolation, and reclassifying grid cells of

excluded land cover types (i.e. water, cropland, urban and built-up, cropland/natural vegetation mosaics, snow and ice, barren

or sparsely vegetated) to missing values.

## 2.3 Data analyses

### 2.3.1 Model fitting and validation

Data analyses focused on the modelling of global $F_{0.99}$ as a function of mean annual $E$ and $D$ (Eq. 2-4) and on the interpretation

of the model in terms of its consistency with current understanding of global fire patterns. Our complete data set consisted of

estimates of mean annual $F$, $P$, $E_0$, and $E$ for 193,476 grid cells covering the selected land cover types of the global land area,

except Antarctica, at 0.25° x 0.25° spatial resolution. A randomly selected sample of 50% of the grid cells was used for model

fitting, while the other 50% of the data was set aside for model validation. We used R (R Core Team, 2018) for all data

analyses, in particular the 'raster' (Hijmans, 2016) and 'quantreg' packages (Koenker, 2017).

Non-linear quantile regression was used to fit Eq (4) to the 0.99 quantile of $F$ as a function of mean annual $E$ and $D$. To

minimize bias in the model towards the most common global climates (e.g. desert or boreal climates), we used a simple

bootstrap procedure of two steps: i) the global $E$, $D$ space was divided into 100 mm by 100 mm bins and all climate bins with

a minimum of 100 grid cells identified (n=145), ii) a random sample (with replacement) of 100 grid cells was drawn from

these bins, and iii) Eq (4) was fitted to the sample data. This procedure was run 1000 times to generate 1000 response surfaces

from which a mean $F_{0.99}$ response surface was calculated (Geyer, 2011). The means and confidence intervals of the fitted

model coefficients are listed in Table 1.

Model predictions were validated against the 50% of the data that was not used for model fitting. To do so the validation data

was binned into 100 mm x 100 mm wide $E$, $D$ bins and the corresponding values of $F_{0.99}$ identified for all bins with a minimum





of 50 observations (N=191). Predicted values of $F_{0.99}$ were extracted from the modelled mean response surface for the same

set of 191 $E, D$ data pairs. Observed values of $F_{0.99}$ correspond to the 0.99 quantile value of $F$ in each of the 191 100 mm x 100 mm $E, D$ bins. The relationship between observed and predicted $F_{0.99}$ was evaluated by linear regression analysis (Supplementary Material, S1). Deviations between observed and predicted $F_{0.99}$ were quantified using the mean difference (MD) and root mean squared difference (RMSD):

$$MD = \sum_{i=1}^{n} (\hat{y}_i - y_i) / n \tag{5}$$

$$RMSD = \sqrt{\sum_{i=1}^{n} (\hat{y}_i - y_i)^2 / n} \tag{6}$$

where $\hat{y}_i$, $y_i$ are predicted and observed values of $F_{0.99}$ and $n$=191. To measure agreement between the spatial patterns of observed and predicted $F_{0.99}$, we used the normalised mean error (NME) and normalised mean squared error (NMSE), as proposed by Kelley et al. (2013):

$$NME = \sum_i |\hat{y}_i - y_i| / \sum_i |y_i - \bar{y}| \tag{7}$$

$$NMSE = \sum_i (\hat{y}_i - y_i)^2 / \sum_i (y_i - \bar{y})^2 \tag{8}$$

where $\hat{y}_i$ is the predicted value of $F_{0.99}$ at grid cell $i$, $y_i$ the corresponding observed value, and $\bar{y}$ the mean of all observed values. By normalising by the spatial variability of the observations, NME and NMSE provide a measure of the spatial error of the model, with NME or NMSE close to 0 indicating perfect agreement between observed and predicted patterns, and both metrics approaching unity when agreement is similar to that of a model that predicts a spatially uniform value equal to the

mean of all observations (Kelley et al., 2013).

### 2.3.2 Identification of climate-fire domains

Following Boer et al. (2016) two climate-fire domains were distinguished on the $P, E_0, F_{0.99}$ response surface depending on whether the direction of the $F_{0.99}$ gradient was more parallel to the local $E$ gradient or $D$ gradient, indicating predominance of

fuel productivity or fuel dryness limitation on fractional burned area, respectively. The boundary between the two climate-fire domains was identified analytically using a gradient analysis of the $F_{0.99}$ response surface relative to gradients of $E$ and $D$ in $P, E_0$ space (Supplementary Material, S3).

As in Boer et al. (2016) we refer to these two domains as productivity-limited (PL) fire and dryness-limited (DL) fire domains and analysed whether the affinity to either domain was related to the vegetation type being dominated by grasses/herbaceous

or woody plants. To this end, areas of homogeneous land cover type were identified on the GlobeLand30 map (Chen et al., 2015). The GlobeLand30 product (Chen et al., 2015) is based on 30 m resolution Landsat imagery and classifies land cover types according to the dominant plant life form (e.g. forest, shrubland and grassland), which can be more readily related to distinct fuel types than biome classifications that often include classes of mixed life forms (e.g. woody savanna in MCD12Q1). A large random sample (N=9,053) of ca. 30 km x 30 km areas of homogeneous land cover type was drawn by sampling 16

blocks of 1000 x 1000, 30 m grid cells from each of the 853 GlobeLand30 tiles and keeping all blocks with at least 75% in a



single land cover class. The geographical coordinates of the 9,053 homogeneous sample blocks were first used to extract corresponding values of $P$ and $E_0$ from the climate grids, which were then used to extract the corresponding domain class from the $P, E_0, F_{0.99}$ response surface.

The distribution of the five global fire regime classes ('pyromes') distinguished by Archibald et al. (2013) over the

productivity-limited (PL) and dryness-limited (DL) fire domains was analysed by: i) mapping all grid cells of the global pyrome map (i.e. Fig. 2 in Archibald et al., 2013) to the $P, E_0, F_{0.99}$ response surface via their corresponding mean annual $P$ and $E_0$, ii) dividing the global $P, E_0$ space in 100 mm x 100 mm bins and identifying the pyrome class with the highest relative frequency in all $P, E_0$ bins with a minimum of 50 data points (N=185), iii) a qualitative evaluation of the consistency of the predicted $F_{0.99}$ and relative importance of fuel production and fuel dryness constraints on fire with the defining characteristics

of the five pyrome classes (Archibald et al., 2013).

## 3 Results

### 3.1 Burned area

According to the GFED4 data base (Giglio et al., 2013), global burned areas amounted to 2.3% of the terrestrial land area per year over the 1995-2016 observation period. The observed mean annual fractional burned area, $F$, was highest in the tropical

savanna regions of Africa, Australia and (to a lesser extent) South America, where $F$ values in the 0.3-0.4 range were common and as high as 0.7-0.8 in localized areas. In other fire-prone environments, $F$ was much lower, with values of up to ~0.10 for shrublands and in the 0.01-0.03 range for most forests.

The 0.99 quantile of the mean annual fractional burned area, $F_{0.99}$, was found to be a highly predictable function of the climatic water balance terms $E$ and $D$, and therefore of mean annual precipitation ($P$) and potential evapotranspiration ($E_0$) (Fig. 1a-c).

Linear regression analysis of observed versus predicted $F_{0.99}$ for validation sites showed that the hydroclimatic model (Eq. 4, Fig. 1c) explained 80% of the global variation in $F_{0.99}$ ($R^2 = 0.80$), with a Mean Difference (MD) between observed and predicted values of 0.05, Root Mean Squared Deviations (RMSD) of 0.14, Normalised Mean Error (NME) of 0.45 and Normalised Mean Square Error (NMSE) of 0.27. Further details on the validation of the $F_{0.99}$ model are provided in Supplementary Material, S1.

The predicted global pattern of $F_{0.99}$ was very similar to the observed pattern of $F$ in the tropical savannas of Sub-Saharan Africa and Australia, where tropical wet-dry climates combine high levels of fuel production during the wet season with intense drought during the dry season, producing $F$ values as high as 0.8-1.0 (Fig. 2a-b). In woody ecosystems outside of the tropical savannas and semiarid grasslands $F_{0.99}$ seldom exceeded 0.5, which is consistent with the fact that most predominantly woody vegetation communities cannot survive such high levels of fire activity over long periods (Bond and Keeley, 2005). Zonal

medians of predicted $F_{0.99}$ were 0.17 for Mediterranean forests, woodlands and scrub, 0.05-0.12 for temperate forests, and 0.01 for boreal forest environments Supplementary Material, S2.





### 3.2 Climate-fire domains

The fitted $P, E_0, F_{0.99}$ response surface consists of two distinct domains (Fig. 1d) characterized by different climate constraints

on fire (Boer et al., 2016). The first domain (green zone in Fig. 1d) is characterized by $F_{0.99}$ increasing with mean annual actual evapotranspiration ($E$) but not with variation in climatic water deficit ($D$), consistent with potential fire activity levels being primarily limited by fuel production (hereafter: PL-type fire). In the second domain (orange zone in Fig. 1d), $F_{0.99}$ increases strongly with $D$ but varies little with increasing $E$, consistent with potential fire activity levels being primarily limited by the capacity of the atmosphere to dry-out fuel material to ignitable levels (DL-type fire).

A visual interpretation of the fitted $P, E_0, F_{0.99}$ response surface suggests that the domain shift from PL-type fire to DL-type fire occurs at some threshold aridity index (i.e. the ratio of mean annual potential evapotranspiration and precipitation: $\frac{E_0}{P}$). The equation for the boundary between the two domains can be derived analytically from Equations (1-4) and is approximately linear for the region covered by the observations: $E_0 = (1.94 \pm 0.02)P$ (p < 0.001 and $R^2 = 0.996$) (Supplementary Material, S3).

Using gridded mean annual climate data, the domains of PL- and DL-type fire were mapped to geographical space (Fig. 2c). The global pattern of PL- and DL-type fire is similar to the global distribution of climate aridity and biome types, as expected given that the boundary between PL- and DL-type fire corresponds to an aridity index of ~1.9. Dryland environments at mid latitudes on all continents were classified in the domain of PL-type fire, while wet and cold environments at all latitudes were classified in the domain of DL-type fire. The domain classification indicates whether the primary limiting factor on mean

annual fire activity levels was fuel production or fuel dryness, which can be expected to correlate strongly with the dominant vegetation lifeform and fuel type (litter versus grass). Using the GLC30 global lifeform mapping (Chen et al., 2015) we found that areas of forest, wetland and tundra were predominantly classified in the domain of DL-type fire (i.e. they are typically too wet to burn for much of the time), whereas grasslands, shrublands and barren lands were predominantly classified in the domain of PL-type fire (i.e. fuels are typically sparse and discontinuous for much of the time) (Fig. 3). The shrubland class is an

interesting case: these ecosystems occur most frequently in the domain of PL-type fire even though they are dominated by woody vegetation and woody plant litter forms an important fuel component. However, classification as PL-type fire makes sense because (semiarid and arid) shrublands can often only support large fires when herbaceous vegetation fills in the gaps and connects the fuel array after above-average rainfall (e.g., O'Donnell et al., 2011;Prior et al., 2017).

As expected, the five pyromes distinguished by Archibald et al. (2013) were found to occupy distinct domains in $P, E_0$ climate

space (Fig. 4): i) the 'RIL' pyrome, characterized by rare, intense, large fires was restricted to low $P$ environments in the climate domain of PL-type fire (green grid cells, Fig. 4) with a median predicted $F_{0.99}$ of 0.05, consistent with its geographical distribution in the more arid zones of boreal forests and temperate coniferous forests, plus areas of Mediterranean vegetation and xeric vegetation; ii) the 'FIL' pyrome, characterized by frequent, intense, large fires (yellow grid cells, Fig. 4), was found



in the climate domain of PL-type fire across a broad range of mean annual $P$ combined with high $E_0$ and with a median
predicted $F_{0.99}$ of 0.55, which is typical for the tropical savannas in Australia and Africa where this pyrome prevails; iii) the
'FCS' pyrome, characterized by frequent, cool (low-intensity), small fires (orange grid cells, Fig. 4), was found across both
climate domains in the region that combines high $P$, high $E_0$, and very high predicted $F_{0.99}$ (median=0.63), corresponding
mainly to tropical grasslands and shrublands as well as tropical dry broadleaf forests; iv) the 'RCS' pyrome, characterized by
rare, cool, small fires (pink grid cells, Fig. 4) was found mostly in the domain of DL-type fire in the region of intermediate $P$
and $E_0$ with the median of predicted $F_{0.99}$ just below 0.09. The geographical distribution of the RCS pyrome spans all
continents and a range of biomes, including large fractions (0.4-0.5) of boreal forests, temperate coniferous and broadleaf
forests, as well as smaller fractions (0.2-0.4) of Mediterranean vegetation and Montane grasslands; v) the 'ICS' pyrome, the
most 'human-driven' pyrome according to Archibald et al. (2013), is characterized by intermediate frequency of cool, small
fires (blue grid cells, Fig. 4) and was found in both climate domains, PL- and DL-type, across wide ranges of $P$ and $E_0$, which
translates to a wide range of predicted $F_{0.99}$ (median=0.25). The geographical distribution of the ICS pyrome is also
widespread, including large fractions (0.4-0.6) of all tropical forest biomes, and smaller but substantial fractions (0.2-0.4) of
temperate forests, (tropical) grasslands and shrublands, which is consistent with the spread across climate domains of both fuel
productivity limitations (PL-type) and fuel dryness limitations (DL-type).

**4 Discussion**

This study has shown that climate sets strong and highly predictable constraints on the global distribution of fire on Earth. In
particular, climate constrains the amounts and timing of plant available water and energy and thereby determines the
probability that the two most basic conditions for fire are met, namely the production and desiccation of plant biomass and
derived fuels. We showed that the strength of those two fundamental climate constraints on fire, and global variation therein,
are captured well by the mean annual climatic water balance, which provides a simple yet biophysically sound basis for the
prediction of potential fractional burned area from just two readily available climate variables: mean annual precipitation and
potential evapotranspiration. The proposed hydroclimatic model was validated against independent burned area data and found
to explain 80% of the global variation in potential fractional burned area ($F_{0.99}$) with a slight tendency to overpredict $F_{0.99}$, but
an NME of 0.45 indicating good agreement between predicted and observed spatial patterns of $F_{0.99}$. A direct comparison of
model performance with existing global fire models is difficult since most existing models predict $F$ rather than $F_{0.99}$ and
systematic evaluation of their performance is ongoing as part of the fire modelling intercomparison project (FIREMIP)
(Hantson et al., 2016;Rabin et al., 2017;Teckentrup et al., 2019;Forkel et al., 2019). Our hydroclimatic model is conceptually
similar to a global model proposed by Kelley et al. (2019) that predicts burned area as a function of four limitations (i.e. fuel
continuity, fuel moisture, potential ignitions and a suppression index). Kelley et al. (2019) report an NME score of 0.60-0.63





for their model predictions of mean annual burned area against the GFED4s data set (Randerson et al., 2012;Giglio et al., 2013), indicating a lack of agreement between the spatial pattern of predicted and observed burned area.

Overprediction of $F_{0.99}$ by our model was somewhat more pronounced in environments of the DL-type fire than in environments of PL-type fire (see Supplementary Material, S1), but model residuals had similar distributions for a wide range of MODIS landcover types, supporting the notion that the model captures the main climatic constraints on global fire activity

levels. Existing fire models tend to struggle predicting the fire regimes of temperate and boreal forest regions, that burn much more infrequently than tropical savannas (Archibald et al., 2013), but we did not observe that model fit suffered in any particular biome because we employed a mechanistic, hydroclimatic approach that captured the fundamental biophysics underlying global fire regimes.

The bias in performance of existing global fire models is likely due to a limited capacity to simulate fuel drying dynamics in

forest and woodland environments. Whereas the seasonally wet-dry climate of tropical savannas makes annual production and subsequent desiccation of fuels highly predictable, fire activity in forests and woodlands is primarily constrained by the moisture content of the (inherently abundant) fuels, which varies at (sub)daily to monthly timescales (Nolan et al., 2016;Caccamo et al., 2012;Boer et al., 2017). The fuel moisture models within several of the existing global fire models (Rabin et al., 2017) are driven by some implementation of the Nesterov index: $N = \sum_{i=1}^{\omega}(T_i - T_{dp,i})T_i$, where $\omega$ is the number of days

since the last rainfall exceeding 3 mm d$^{-1}$, $T_i$ is the daily 3 pm air temperature (°C) and $T_{dp,i}$ the daily 3pm dew point temperature (°C). As shown by Schunk et al.(2017) the Nesterov Index was a poor predictor of the moisture content of 1-hour and 10-hour fuels from four main forest species in Germany. With fuel moisture variation modelled by the Nesterov index global fire models lack the critical capacity to predict when, or how frequently, forests and woodlands switch from a moist/non-flammable state to a dry/flammable state and are therefore unlikely to reproduce observed spatiotemporal variation in burned

area in those biomes. In contrast, our hydroclimatic model does not predict fuel moisture content; instead, the mean annual climatic water deficit ($D$) is used as an estimate of the probability of extant fuels drying out to ignitable levels during some fraction of the year. With $D$ being a measure of the atmospheric moisture demand that cannot be met by the soil water store (Stephenson, 1998), $D$ captures the basic biophysics involved in the desiccation of fuels and accounts for the fact that sparse fuels (low $E$) require less energy ($E_0$) to dry out than dense/heavy fuels.


Further evidence for $D$ being a reasonable indicator of the mean annual probability of forests and woodlands being in a dry fuel state can be derived from previous studies on climate-fire relations in the southwest Unites States (Williams et al., 2015;Abatzoglou et al., 2017) and studies on dead fuel moisture (Resco de Dios et al. 2015) and fire activity in temperate forests of SE Australia (Nolan et al., 2016) and Portugal (Boer et al., 2017) that showed that cumulative burned area in these

different forest regions responds strongly non-linearly to predicted fine dead fuel moisture content dropping below thresholds identified at 14-18% and 10-12%, respectively. Using the Resco de Dios et al. (2015) fuel moisture model with gridded global vapour pressure deficit data to predict daily fine dead fuel moisture content for global forests and woodlands, we found that





mean annual $D$ is strongly and linearly related (adj. $R^2$: 0.76, $p<2e-16$) to the mean annual frequency of predicted daily fine dead fuel moisture content dropping below 10% (Supplementary Material, S4).

The hydroclimatic model allowed us to objectively distinguish climatic domains for a predominance of either fuel productivity (PL-type) or fuel dryness (DL-type) constraints on mean annual fractional burned area, and showed their geographical distribution to be consistent with global patterns of herbaceous vs. woody vegetation types and corresponding fuel types, in accordance with the variable constraints hypothesis (Krawchuk and Moritz, 2011). Further, we demonstrated (see Supplementary Material, S3) that the boundary between the domains of PL- and DL-type fire is well approximated by an

aridity index of 1.9 (1.94±0.02), providing the first objective identification of where in climate space (Fig. 1d), and in geographical space (Fig. 2c), fire regimes switch from fuel load limitations to fuel moisture limitations. We showed that the hydroclimatic model and associated classification in two contrasting climate-fire domains is consistent with the hydroclimatic distribution of pyromes (Archibald et al., 2013), indicating that key aspects of global fire regimes vary in a predictable way with global gradients in mean annual precipitation and potential evapotranspiration.

By focusing exclusively on the roles of fuel production and fuel dryness constraints, our hydroclimatic model was designed to predict the potential or maximum mean annual fractional burned area, which provides a useful reference for bottom-up process-based modelling approaches used in many DGVMs (Hantson et al., 2016). Further work should be extended to predicting annual, rather than potential mean annual, fractional burned area by modelling the deviations between predicted $F_{0.99}$ and observed annual fractional burned area as a function of fire weather and ignition constraints on fire activity, thus completing

the formalisation and implementation of the four-switch concept (Bradstock, 2010). Other future work could also examine the drivers, such as fire management and other human activities, of geographical variation between $F$ and $F_{0.99}$.

**5 Conclusion**

At long time scales the global distribution of fire on Earth is highly predictable from fundamental biophysical constraints on

the production and seasonal desiccation of plant biomass (i.e. fuel), which in turn are proportional to mean annual precipitation and potential evapotranspiration. The sharp transition of global fire regimes from domains of fuel production limitations on fire (PL-type) to fuel dryness limitations on fire (DL-type) can be identified from the mean annual aridity index being above or below a threshold value of 1.9. Our model provides a simple but comprehensive basis for predicting fire potential under current and future climates, as well as an overarching framework for estimating effects of human activity via ignition regimes

and manipulation of vegetation. In these respects, it offers a significant advance on existing global fire models and therefore a basis for improving predictions from coupled global vegetation models.

**Data availability.**

Global Fire Emissions Database (GFED) data are freely available at: http://www.globalfiredata.org/.





WorldClim gridded climate data can be downloaded from: http://worldclim.org/.

Gridded mean annual global potential evapotranspiration can be downloaded from CGIAR-CSI: https://cgiarcsi.community/data/global-aridity-and-pet-database/.

The Globeland30 (GLC30) global landcover classification layer can be downloaded from: http://www.globallandcover.com/.

The Moderate Resolution Imaging Spectroradiometer (MODIS) landcover classification (MCD12Q1) can be downloaded
from: https://lpdaac.usgs.gov/dataset_discovery/modis/modis_products_table/mcd12q1.

The global pyrome classification layer is available upon request from Sally Archibald.

**Competing interests.** The authors declare that they have no conflict of interest.

**Author contributions.** M.M.B. conceptualised the study, performed the data analyses and wrote the initial draft. E.Z.S. performed the gradient analysis. V.R.D. and R.A.B. contributed to the interpretation of results and the writing of the final manuscript.

**Acknowledgements.** M.M.B and R.A.B. acknowledge funding support from the New South Wales Office of Environment
and Heritage through the NSW Bushfire Risk Management Research Hub. E.Z.S. acknowledges support from the Australian Research Council (DP160103436). We thank Sally Archibald for providing the pyrome classification layer.

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





**Tables**

**Table 1. Mean and 95% confidence intervals of model coefficients (Eq. 4) obtained from 1000 model fits.**

| Coefficient | Dimension | Mean | 95% CIs |
|:---:|:---:|:---:|:---:|
| $E_1$ | mm.y$^{-1}$ | 577 | 554, 597 |
| $E_2$ | mm.y$^{-1}$ | 976 | 952, 1126 |
| $D_1$ | mm.y$^{-1}$ | 607 | 591, 626 |
| $D_2$ | mm.y$^{-1}$ | 1034 | 986, 1088 |


**Figures**

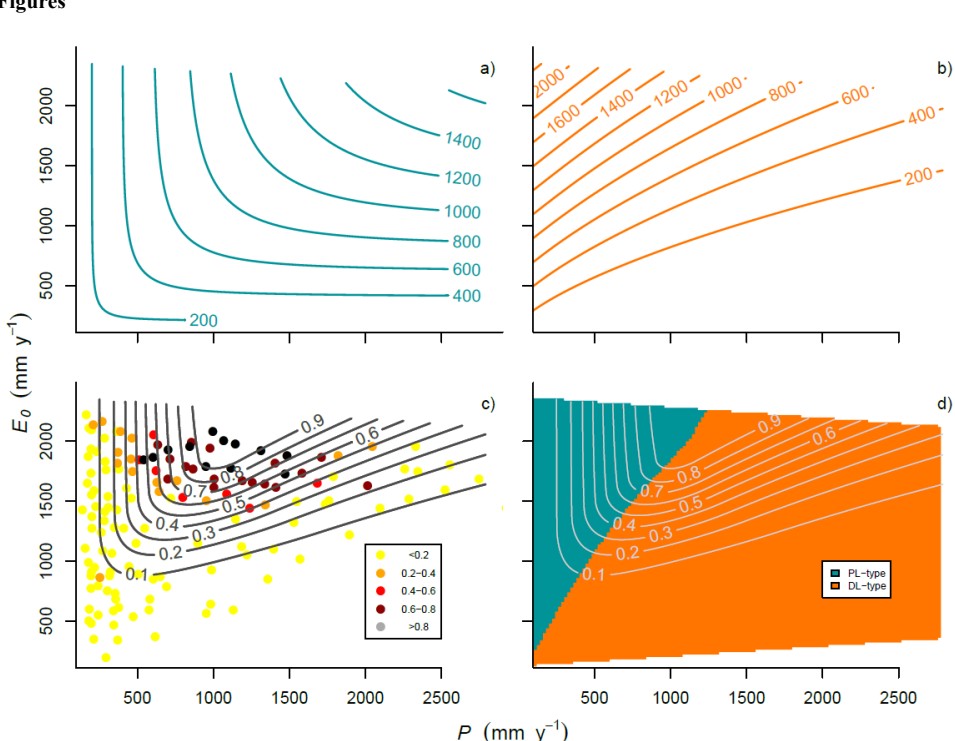

**Figure 1. a) Mean annual actual evapotranspiration (*E*, mmy-1), b) climatic water deficit (*D*, mmy-1), and c) the 0.99 quantile of mean annual fractional burned area ($F_{0.99}$) as functions of mean annual precipitation (*P*) and potential evapotranspiration ($E_0$). d) Distribution of the domains of PL- and DL-type fire in *P*, $E_0$ space, with contours of the $F_{0.99}$ model in grey. Coloured dots in panel c) correspond to independent observations of $F_{0.99}$ used for validation.**

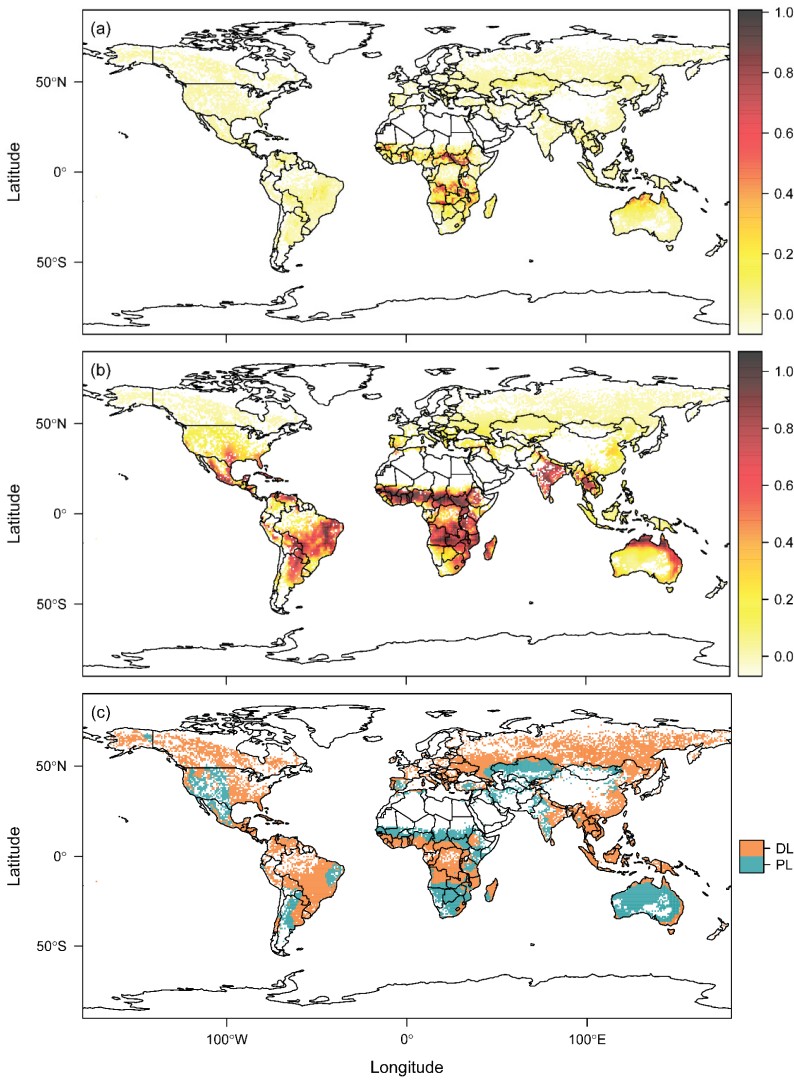

**Figure 2. a) Observed (1995-2016) mean annual fractional burned area ($F$). b) Predicted 0.99 quantile of mean annual fractional burned area ($F_{0.99}$). c) Geographical distribution of domains of PL- and DL-type fire in green and orange tones, respectively. In white land areas the observed mean annual fractional burned area was negligible ($F < 0.000001$).**



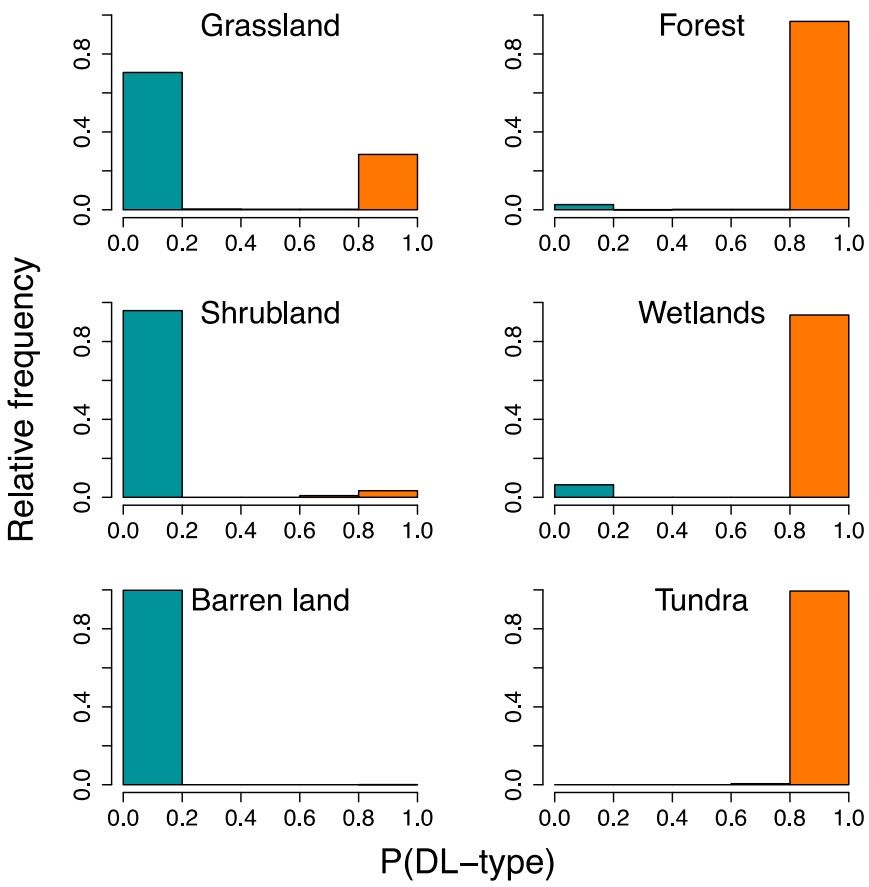

**Figure 3. Distributions of probabilities of classification in the domain of DL-type fire for a large sample (N=9053) of 30 km x 30 km areas of homogenous landcover drawn from the GLC30 global lifeform database (Chen et al., 2015). Bars are coloured green where P(DL-type)<0.5, to indicate prevailing affinity to the domain of PL-type fire, and coloured orange where P(DL-type)>0.5 to indicate prevailing affinity to the domain of DL-type fire.**



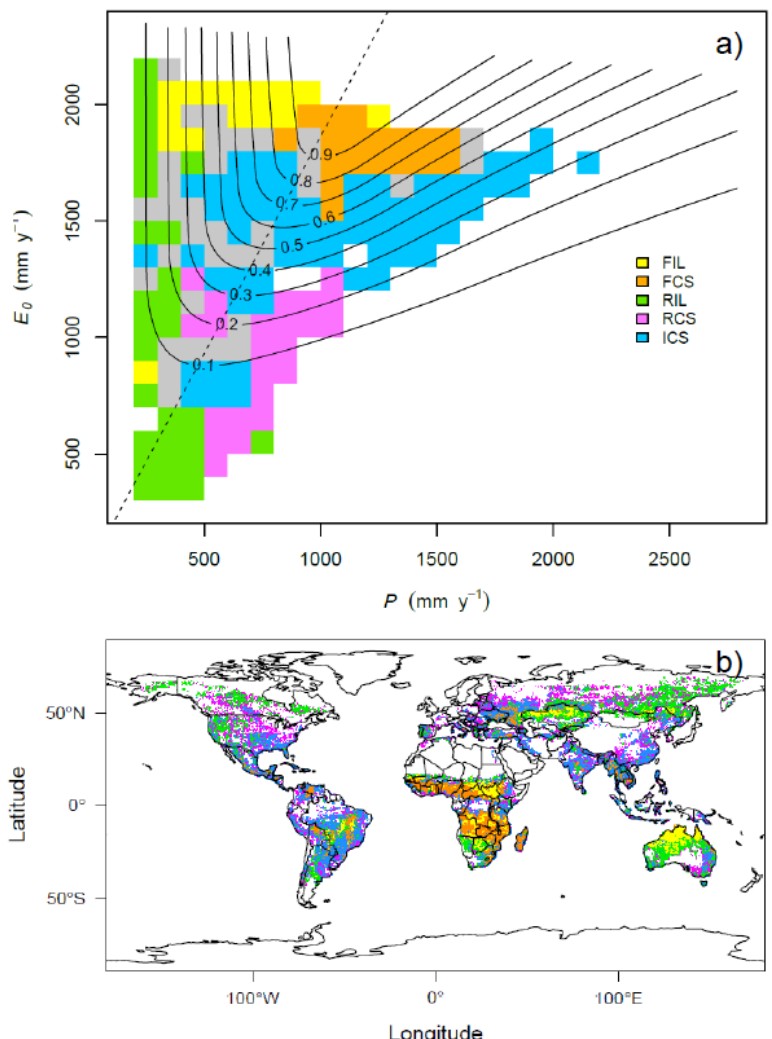

**Figure 4. a) Distribution of global pyrome classes (Archibald et al., 2013) in $P, E_0$ space, with contours of the fitted global $F_{0.99}$ model in black and grey cells indicating climate space where none of the five pyromes had a relative frequency exceeding 0.2. b) Geographical distribution of pyrome classes, redrawn from Archibald et al., 2013.**
