# Peer review of "A hydroclimatic model for the distribution of fire on Earth"

_Biogeosciences, 2019_

## Referee Comment (RC1) · Anonymous Referee #1 · 19 Jan 2020

General Comments: The authors present a new approach for modeling a measure of fire activity (maximum annual burned area) using a water-balance approach. Through the use of two variables that are linked through energy-water dynamics, they do a nice job using E (a measure of productivity) and D (a measure of drought stress) in elucidating fuel and flammability limited fire regimes globally under late 20th-century climate. There are a few areas where the work is weak and can be improved upon that I provide major considerations for below:

Major Considerations

1) I struggled to get a firm handle on the interpretation of F99. I believe it represents some maximum mean annual burned area, but I may be interpreting it (in my head) as the maximum burned area in one year. If referring to the former, it is possible that

the results may impact the use of the Budyko curve. The Budyko curve partitioning E and D from P/PET; in areas with very high mean annual burning one might expect substantial departures from the Budyko curve as frequent fire would strongly shape both biomass abundance and soil properties. Better clarification on the exact variable of interest and why it is of value in global fire analysis would be helpful to myself and likely other readers.

2) The Pyromes data from Archibald et al. doesn't seem to provide much value in the current study. There are problems with the pyrome layer that make it hard to understand (e.g., constructed from a short record and biased by the spatial unit of analysis where you have adjacent pixels with similar veg/climate that get classified to different pyromes); do the authors think that maximum annual burned area should be part of what defines a pyrome, or through the lens of those defined? If not, it is a bit of a fishing expedition. For example, in Figure 4 there are strange non-contiguous regions (e.g., ICS spanning huge region both fuel and flammability-limited). I believe this is a limitation/problem with the fire regimes, not the methodology of the current paper. Since there are no very strong arguments to expect F99 to map onto these pyromes, I would suggest analyses relating to pyromes to not be necessary.

3) The Budyko curve is a generalization of the partitioning of precipitation into runoff and actual evapotranspiration. However, there are numerous studies that show that there are substantial deviations from this curve that materialize due to vegetation, soil water holding capacity, the seasonal synchronicity of P and PET, and precipitation phase. At the very least it is worth acknowledging this and how it may impact estimates of E. reference to this as it will impact your estimates of E. At the most, the authors may also consider using gridded modeled estimate of E and D that are available globally at spatial resolutions sufficient for the penultimate scale of GFED.

Minor considerations

4) Line 33, This sentence is unclear "The partial success in current models. . .", please

rephrase or clarify.

5) Eq 2 & 3, How do your results compare using these parametric equations to a completely non-parametric approach?

6) Line 105, The GFED data for the first several years of record is highly suspect due to inhomogeneities in data source. I would repeat the analysis using only GFED from the MODIS-only era. Also, do your results differ if you include small fires GFED (GFED 4s?)

7) Line 114: I am unaware of WorldClim data covering this entire period, v1 is 1961-1990 and v2 is 1971-2000; The documentation for the Eo data says 1950-2000, but I wonder if that is a misinterpretation as there is no WorldClim for this period.

8) Line 120: Note that using the GFED database you can disaggregated burned area by MODIS land cover class and excluded agricultural burns. This might be a cleaner approach as there are many assumptions with interpolating MODIS land cover. Also note that there exist MODIS land cover classifications at 0.25 degree resolution to match GFED [https://ldas.gsfc.nasa.gov/gldas/vegetation-class-mask]

9) Line 223: A distinction here is that this approach clearly defines tundra as PL-type; whereas vegetation assembles or other fire regime classifications might define this as fuel limited.

10) Line 277: Is there a way to demonstrate that model skill does not degrade as a function of fire return interval, etc?

11) Line 301: The numbers "14-18% and 10-12%, respectively" are hard to follow. Since I do not think the paper relies on them, I might omit them here but keep the citations.

12) Figure 1: legend: replace grey dot with black

13) Figure 2: I think this is stated in the discussion and probably a place for follow-up

work, but it would be clearly of strong value to parse out how other factors (e.g., human footprint, lightning density, etc) explain the variance between F99 and F. Since we are assuming the climate factors shape F99, the hope is that non-climatic factors are not strongly influencing the pixels where observed F99 occurs.

14) There is some similarity to the approach here and the empirical approach of Guyette et al. (2012), although they use fire histories and try to estimate mean fire return interval

Guyette, R.P., Stambaugh, M.C., Dey, D.C. and Muzika, R.M., 2012. Predicting fire frequency with chemistry and climate. Âǎ Ecosystems, Âǎ 15(2), pp.322-335.

---

## Referee Comment (RC2) · Anonymous Referee #2 · 10 Mar 2020

Review for Boer et al: A hydroclimatic model for the distribution of fire on Earth Summary: The authors present a hydroclimatic model to estimate potential maximum burnt area across climate space. They link the outcome of this model to fuel vs moisture limited fire regimes. Overall, this is a nice and coherent manuscript presenting interesting results using a sound methodology. The authors can find some minor comments below, which might help to further improve and/or clarify the manuscript.

Main comments

The interpretation of F99 is a bit hard to imagine. I kind of interpret it as the maximum an area can potentially burn considering the mean climatic conditions, but it would be nice if the authors could indicate what they think is the best interpretation of F99 so that the reader doesn't need to imagine this. This would especially help for people who

don't have the time to read the methods section to understand what F99 actually is (when one starts reading it is confusing what F99 actually is).

Related to this topic, I think the discussion covers some interesting topics, but I miss some depth in how we could use this F99 estimate to improve our understanding of the drivers of global fire activity beyond the results of the paper. E.g. difference between F and F99 can indicate human impact, but can also differences in vegetation type, structure and traits under similar climate conditions, and can possibly guide us to explain some of the continental differences observed in burnt area (e.g. Lehmann et al., 2014).

I think the methods are sound and the results overall robust. However, I have some doubt for areas with very long fire return intervals such as the boreal region. In these areas, large fires result in very high burn fractions within a 0.25° gridcell for a given year, which you then divide by the length to the time series to get your F. However, doesn't this mean that your F99 will depend on the length of the time series used for these regions with long fire return intervals and large fire sizes? I point this out because you use the GFED data from 1995, but the pre-MODIS data is much less reliable, so I would suggest to only use the MODIS era data (just a suggestion if it is not too much trouble).

When looking globally at burnt area, and especially at extremes, one tends to only see Africa, which has so much burnt area than any other continent that it tends to completely dominate any analysis. In your methods you implement a bootstrapping, but I do wonder how different your F99 estimates would look without Africa (or the other direction, how much does it matter to include the rest of the world in the analysis?). This is always a problem, and no criticism, but for interpretation of the results it could be nice to know this kind of "uncertainty".

Minor comments

L8: maybe add human, as population density is supposed to be a good indicator of ignitions and suppression.

L12-13: 99 percentile over? Time/space? It should be explicitly indicated how this is calculated. After reading the methods section and your previous manuscript over Australia I notice that it is a 99 percentile quantile regression, you should make this clearer in the sections which come before the methods (and possibly even for the results for people who don't want to go over the methods to interpret the results).

L79-80: P and Eo are not yet defined.

L111-113: Does it matter that P and Eo come from different sources, e.g. a physical disconnection between both could influence your estimates for D?

L212-214: I think this separation between production and dryness is nice. However, shouldn't there be a precipitation level where the default it is fuel limited? E.g. NPP is very low and almost exclusively limited by precipitation under very dry conditions. Now it seems that under even very low precipitation values it can you still be moisture limited?

L275: For a quantification of the spatial uncertainty in fire models you might be interested in this recent paper: https://www.geosci-model-dev-discuss.net/gmd-2019-261/

L290-294: I do follow the logic in using D for multiannual mean fire conditions (and I agree that the Nesterov Index is suboptimal), I don't see the logic in comparing predictions which are generally made at (sub)daily timesteps to your multiannual average F(99) estimates. These seem to be two pretty disconnected things and I don't see how you could use D for these short timestep responses in burnt area.

L310: Another comment about regarding this dichotomy between fuel and dryness, there was a recent paper by Alvarado et al which investigated this across the tropics and found important differences between continents, but only looking at precipitation. I wondered whether these results could explain these differences by looking to over an aridity gradient?

References

Alvarado, ST, Andela, N, Silva, TSF, Archibald, S. Thresholds of fire response to moisture and fuel load differ between tropical savannas and grasslands across continents. Global Ecol Biogeogr. 2020; 29: 331– 344. https://doi.org/10.1111/geb.13034

Lehmann, C. E. R., Anderson, T. M., Sankaran, M., Higgins, S. I., Archibald, S., Hoffmann, W. A., Hanan, N. P., Williams, R. J., Fensham, R. J., Felfili, J., Hutley, L. B., Ratnam, J., San Jose, J., Montes, R., Franklin, D., Russell-Smith, J., Ryan, C. M., Durigan, G., Hiernaux, P., Haidar, R., Bowman, D. M. J. S., and Bond, W. J.: Savanna Vegetation-Fire-Climate Relationships Differ Among Continents, Science, 343, 548-552, 10.1126/science.1247355, 2014.

---

## Author Comment (AC1) · 16 Apr 2020

General Comments: The authors present a new approach for modeling a measure of fire activity (maximum annual burned area) using a water-balance approach. Through the use of two variables that are linked through energy-water dynamics, they do a nice job using E (a measure of productivity) and D (a measure of drought stress) in elucidating fuel and flammability limited fire regimes globally under late 20th-century climate. There are a few areas where the work is weak and can be improved upon that I provide major considerations for below:

Thank you for your constructive comments (black font). Below we have added our responses to each individual comment in blue font.

Major Considerations
1) I struggled to get a firm handle on the interpretation of F99. I believe it represents some maximum mean annual burned area, but I may be interpreting it (in my head) as the maximum burned area in one year. If referring to the former, it is possible that the results may impact the use of the Budyko curve. The Budyko curve partitioning E and D from P/PET; in areas with very high mean annual burning one might expect substantial departures from the Budyko curve as frequent fire would strongly shape both biomass abundance and soil properties. Better clarification on the exact variable of interest and why it is of value in global fire analysis would be helpful to myself and likely other readers.

**Response:** We appreciate that readers may not be familiar with quantile regression and have therefore added a citation of an introductory paper on the topic (Cade et al. 2003. https://doi.org/10.1890/1540-9295(2003)001[0412:AGITQR]2.0.CO;2) in Line 65 where we introduce $F_{0.99}$.

In the introduction we defined $F$ as the mean annual fractional burned area. This is computed from the cumulative burned area recorded over the 1995-2016 GFED observation period, divided by the number of observation years (i.e. 20). $F_{0.99}$ is the 0.99 quantile value of $F$. We modelled $F_{0.99}$ as a function of two hydroclimatic variables: mean annual precipitation, $P$, and potential evapotranspiration, $E_0$. $F_{0.99}$ can thus be interpreted as the maximum or potential value of the mean annual fractional burned area ($F$) for a given set of hydroclimatic conditions. Therefore, $F_{0.99}$ is a prediction of the potential mean annual burned area as a fraction of a 0.25° x 0.25° grid cell and does not represent the maximum [fractional] burned area in one year.

In our view, the global model of $F_{0.99}$ contributes to the foundation of pyrogeography by providing a simple yet robust prediction of global variation in mean annual burned area as a function of the two most fundamental environmental limitations on fire, namely the availability of energy and water for the production and (seasonal) desiccation of fuels.

The Budyko curve provides a prediction of the mean annual partitioning of precipitation in streamflow and evaporation for large catchments or land areas as a function of the aridity index ($E_0/P$). Indeed, many variations of the original Budyko curve have been proposed to account for effects of, for example, climate seasonality, vegetation and soil types, and topography on the long-term water balance. Several studies have evaluated the effect of fire on catchment water balances using the Budyko framework (e.g. Roderick and Farquhar, 2011. doi:10.1029/2010WR009826) but we are not aware of any studies that have

incorporated effects of particular fire regimes in the Budyko framework. However, as shown in our study and many others, very high mean annual fractional burned area is limited to seasonally wet/dry (sub)tropical climates that combine high fuel production rates with strong desiccation of fuels in the dry season. In these highly seasonal environments, we don't expect a strong effect of the mean annual burned area fraction on the mean annual partitioning of precipitation in streamflow and evaporation, because the fires burn in the dry season and mainly affect dry grassy fuels that grow back in the following wet season.

2) The Pyromes data from Archibald et al. doesn't seem to provide much value in the current study. There are problems with the pyrome layer that make it hard to understand (e.g., constructed from a short record and biased by the spatial unit of analysis where you have adjacent pixels with similar veg/climate that get classified to different pyromes); do the authors think that maximum annual burned area should be part of what defines a pyrome, or through the lens of those defined? If not, it is a bit of a fishing expedition. For example, in Figure 4 there are strange non-contiguous regions (e.g., ICS spanning huge region both fuel and flammability-limited). I believe this is a limitation/problem with the fire regimes, not the methodology of the current paper. Since there are no very strong arguments to expect F99 to map onto these pyromes, I would suggest analyses relating to pyromes to not be necessary.

**Response:** We do not suggest that $F_{0.99}$ should be incorporated in the pyrome classification. This would not make sense as the pyrome classification is based on actually observed fire regime metrics, while our predicted $F_{0.99}$ is a prediction of the potential mean annual fractional burned area for a given combination of mean annual precipitation and potential evapotranspiration.
However, we do believe there is interest in exploring how current fire regime syndromes (i.e. pyromes) map to the hydroclimatically defined domains of PL- and DL-type fire and to the predicted potential mean annual burned area ($F_{0.99}$). We expected most of the pyrome classes to be limited to either the PL domain or DL domain, and the range of predicted $F_{0.99}$ to be consistent with the observed values of annual burned area and fire return interval for each pyrome. As presented in the last paragraph of our Results section (L. 249-268) and visualized in Figure 4, three out of five pyromes (i.e. FIL, RIL, RCS) predominantly occupy one of the two domains in accordance with Archibald et al's description, while the pyrome with the strongest human influence according to Archibald et al. (ICS) occurs across both PL and DL domains. The FCS pyrome, which is observed in tropical and temperate grasslands as well as a range of tropical forest biomes (see Table 1 in Archibald et al.), is also spread across the two domains, as expected.

3) The Budyko curve is a generalization of the partitioning of precipitation into runoff and actual evapotranspiration. However, there are numerous studies that show that there are substantial deviations from this curve that materialize due to vegetation, soil water holding capacity, the seasonal synchronicity of P and PET, and precipitation phase. At the very least it is worth acknowledging this and how it may impact estimates of E. reference to this as it will impact your estimates of E. At the most, the authors may also consider using gridded modeled estimate of E and D that are available globally at spatial resolutions sufficient for the penultimate scale of GFED.

**Response:** The Budyko framework is widely accepted as a reasonable global model for the partitioning of precipitation in runoff and evapotranspiration for large catchments or land areas (e.g. Wang et al., 2016, DOI: 10.1177/0309133315620997). It thus provides a reasonable approach to modelling global variation in mean annual actual evapotranspiration from two

input variables, mean annual precipitation and potential evapotranspiration, that are available globally at adequate spatial resolution. We acknowledge the point that local climate, soil, vegetation and terrain can cause mean annual actual evapotranspiration to deviate from the estimate predicted by the Budyko curve. We do not believe we introduced any particular bias in our hydroclimatic model of global fire patterns by using the Budyko curve to predict mean annual actual evapotranspiration. Such deviations are to be expected for any highly simplified global model, but discussion of underlying causes and/or proposing alternative forms of the Budyko curve are beyond the scope of our study. We now provide a description of the terms in equation 1 and refer (L. 91-102) to the abovementioned paper by Wang et al (2016) as a source for further details on the assumptions underlying the Budyko framework, as well as a review of applications in hydrology and hydroclimatology.

Minor considerations
4) Line 33, This sentence is unclear "The partial success in current models: : :", please rephrase or clarify.
**Response:** We have substituted "The partial success of current models" to "The limited ability of current models" (L. 35) to refer to the statement in the previous sentence that current models can reproduce observed fire activity patterns in some environments but not in others.

5) Eq 2 & 3, How do your results compare using these parametric equations to a completely non-parametric approach?
**Response:** Of course, there are alternative models/equations to describe the relationship between fractional burned area and (hydro-)climatic predictor variables. We opted for a parametric model because we know from theory and previous work what shape the relationship between burned area and E or D is likely to have. In our previous study on Australian fire regimes (Boer et al. 2016, doi.org/10.1088/1748-9326/11/6/065002) we showed that a logistic or other sigmoidal function provides good fits for the effects of E and D on burned area; the form of the logistic model is consistent with the 4-switch concept underlying our hydroclimatic model (i.e. simulating a threshold response in burned area once E or D exceeds a given level, i.e. a limiting factor is overcome. The logistic model proposed by Yin et al. 2003 provides a good level of flexibility with two parameters that can be interpreted in biophysical terms of how climate constrains fire activity.

6) Line 105, The GFED data for the first several years of record is highly suspect due to inhomogeneities in data source. I would repeat the analysis using only GFED from the MODIS-only era. Also, do your results differ if you include small fires GFED (GFED 4s?)
**Response:** The GFED4 data base has been used for numerous studies on global fire, including (modelling) studies on climate-fire relationships (e.g. Abatzoglou et al. 2018, Global Change Biology, https://doi.org/10.1111/gcb.14405). The majority of the data base (2000-2016) was built from MODIS burned area products, which was extended with 5 years into the pre-MODIS era (1995-1999) by calibrating monthly active fire counts from the VIRS and ATSR sensors to MCD64A1 burned area data (Giglio et al. 2013, Biogeosciences 118). We have used the GFED4 data for its intended purpose as stated by Giglio et al. (2013): "As with previous versions of GFED, the data set is primarily intended for use within large-scale atmospheric and biogeochemical models and for interpreting regional and continental-scale controls on fire activity from climate and different forms of land management".
Finally, since we are not modelling temporal trends in burned area, but model mean annual burned area, potential errors resulting from the combination of more than one data source in GFED4 are likely to be small.

We did not use GFED4s, so cannot compare with GFED4 results. We note that including or excluding small fires is likely to be particularly important in agricultural areas, which were excluded from our analyses (see section 2.2.3.).

7) Line 114: I am unaware of WorldClim data covering this entire period, v1 is 1961- 1990 and v2 is 1971-2000; The documentation for the Eo data says 1950-2000, but I wonder if that is a misinterpretation as there is no WorldClim for this period.
**Response:** We used WorldClim1 for gridded mean annual precipitation, which according to Hijmans et al., 2005 (DOI: 10.1002/joc.1276) is based on a combination of data bases from the 1950-2000 period.
According to Zomer et al. 2008 (doi:10.1016/j.agee.2008.01.014), the CGIAR-CSI gridded mean annual potential evapotranspiration layer is based on 1960-1990 data. We have corrected section 2.2.2., L. 125-126 accordingly.

8) Line 120: Note that using the GFED database you can disaggregated burned area by MODIS land cover class and excluded agricultural burns. This might be a cleaner approach as there are many assumptions with interpolating MODIS land cover. Also note that there exist MODIS land cover classifications at 0.25 degree resolution to match GFED [https://ldas.gsfc.nasa.gov/gldas/vegetation-class-mask]
**Response:** We applied the MCD12Q1 Land Cover Type classification to exclude several non-native vegetation land cover classes as specified in section 2.2.3.

9) Line 223: A distinction here is that this approach clearly defines tundra as PL-type; whereas vegetation assembles or other fire regime classifications might define this as fuel limited.
**Response:** As shown in Figure 3 (which is the result of 1000 model fits and classification of $P$, $E_0$ space into domains of PL- and DL-type fire) homogenous areas of tundra vegetation were observed to have very high probability of classification in the domain of DL-type. The bar graph shows that probability, i.e. P(DL-type), to be mostly in the 0.8-1.0 interval. This is consistent with the wet conditions prevailing in many tundra environments. Figure 3 was created

10) Line 277: Is there a way to demonstrate that model skill does not degrade as a function of fire return interval, etc?
**Response:** Our model does not predict fire return intervals but potential mean annual fractional burned area ($F_{0.99}$), which can be interpreted as a measure of the multiplicative inverse of the (minimum) fire return interval. We have added a plot to Supplementary Material S1 of the standardized $F_{0.99}$ model residuals versus the fitted $F_{0.99}$ values; it shows that model skill does not degrade with predicted $F_{0.99}$.

11) Line 301: The numbers "14-18% and 10-12%, respectively" are hard to follow. Since I do not think the paper relies on them, I might omit them here but keep the citations.
Response: We specify the threshold values in predicted dead fuel moisture content (DFMC) from the cited studies to support our choice of using a DFMC=10% threshold to show that mean annual climatic water deficit is strongly related to the mean annual frequency of predicted daily DFMC dropping below 10% (Supplementary Material S4)

12) Figure 1: legend: replace grey dot with black
**Response:** Thank you. The grey dot has been replaced with a black dot.

13) Figure 2: I think this is stated in the discussion and probably a place for follow-up work, but it would be clearly of strong value to parse out how other factors (e.g., human footprint, lightning density, etc) explain the variance between F99 and F. Since we are assuming the climate factors shape F99, the hope is that non-climatic factors are not strongly influencing the pixels where observed F99 occurs.

**Response:** As explained in the introduction, our hydroclimatic model builds on the 4-switch concept (Bradstock, 2010. doi.org/10.1111/j.1466-8238.2009.00512.x) and is designed to capture the long-term effects of the two primary biophysical constraints on mean annual fractional burned area, fuel production and fuel desiccation. In this conceptual framework, the frequency/density of ignitions (e.g. through management) and frequency of favourable fire weather conditions (or fire suppression) add further constraints on mean annual burned area, i.e. they reduce burned area below the potential ($F_{0.99}$) set by the primary constraints of fuel production and desiccation. We therefore assume that in areas where the predicted $F_{0.99}$ is close to the observed mean annual fractional burned area ($F$), other constraints than fuel production and fuel desiccation are not limiting mean annual burned area.

14) There is some similarity to the approach here and the empirical approach of Guyette et al. (2012), although they use fire histories and try to estimate mean fire return interval

Guyette, R.P., Stambaugh, M.C., Dey, D.C. and Muzika, R.M., 2012. Predicting fire frequency with chemistry and climate.ĂˇaEcosystems,Ăˇa15(2), pp.322-335.

**Response:** Indeed, Guyette et al. show that key aspects of continental fire regimes can be explained from first principles of physical chemistry. Pointing out similarities with our hydroclimatic model would go beyond the scope of our paper.

---

## Author Comment (AC2) · 16 Apr 2020

Review for Boer et al: A hydroclimatic model for the distribution of fire on Earth Summary: The authors present a hydroclimatic model to estimate potential maximum burnt area across climate space. They link the outcome of this model to fuel vs moisture limited fire regimes. Overall, this is a nice and coherent manuscript presenting interesting results using a sound methodology. The authors can find some minor comments below, which might help to further improve and/or clarify the manuscript.

Thank you for your constructive comments (black font). Below we have added our responses to each individual comment in blue font.

Main comments
The interpretation of F99 is a bit hard to imagine. I kind of interpret it as the maximum an area can potentially burn considering the mean climatic conditions, but it would be nice if the authors could indicate what they think is the best interpretation of F99 so that the reader doesn't need to imagine this. This would especially help for people who don't have the time to read the methods section to understand what F99 actually is (when one starts reading it is confusing what F99 actually is).
**Response:** In the introduction (L. 62-75) we defined $F$ as the mean annual fractional burned area. That is, the cumulative burned area recorded over the 1995-2016 GFED observation period divided by the number of observation years (i.e. 20). $F_{0.99}$ is the 0.99 quantile value of $F$. We modelled $F_{0.99}$ as a function of two hydroclimatic variables: mean annual precipitation, $P$, and potential evapotranspiration, $E_0$. $F_{0.99}$ can thus be interpreted as the maximum or potential value of the mean annual fractional burned area ($F$) for a given set of hydroclimatic conditions. We also added a reference (L. 65) to an introductory paper on quantile regression and its applications in ecology (Cade et al. 2003)

Related to this topic, I think the discussion covers some interesting topics, but I miss some depth in how we could use this F99 estimate to improve our understanding of the drivers of global fire activity beyond the results of the paper. E.g. difference between F and F99 can indicate human impact, but can also differences in vegetation type, structure and traits under similar climate conditions, and can possibly guide us to explain some of the continental differences observed in burnt area (e.g. Lehmann et al., 2014).
**Response:** Our model is built on the 4-switch concept (Bradstock, 2010. doi.org/10.1111/j.1466-8238.2009.00512.x), which describes biogeographical variation in fire activity as a function of a hierarchy of fundamental constraints (i.e. the 4 switches or limiting factors). We focused on the two primary constraints, fuel production and desiccation, and proposed a hydroclimatic work to model their effect on global burned area patterns. We demonstrated that climatic constraints on fuel production and desiccation explain circa 80% of global variation in potential mean annual burned area.
As mentioned in our conclusions, the predicted $F_{0.99}$ and the difference between $F_{0.99}$ and $F$ provide useful starting points for investigating the extent to which human activity could alter fire activity levels by manipulating ignition regimes or vegetation. This will be subject of future studies.

I think the methods are sound and the results overall robust. However, I have some doubt for areas with very long fire return intervals such as the boreal region. In these areas, large fires result in very high burn fractions within a 0.25_ gridcell for a given year, which you then divide by the length to the time series to get your F. However, doesn't this mean that your F99 will depend on the length of the time series used for these regions with long fire return intervals and large fire sizes? I point this out because you use the GFED data from 1995, but the pre-MODIS data is much less reliable, so I would suggest to only use the MODIS era data (just a suggestion if it is not too much trouble).

**Response:** The reviewer suggests that the estimates of the mean annual fractional burned area ($F$), or of the 0.99 quantile of $F$ ($F_{0.99}$), in environments characterized by rare but large fires, and long fire return intervals may depend on the length of the observation period.

We agree that the uncertainty in estimates of $F$ and $F_{0.99}$ for any particular grid cell will decrease with the length of the observation period, in particular for grid cells with relatively long fire return intervals.

However, our hydroclimatic model predicts $F_{0.99}$ for combinations of mean annual precipitation ($P$) and potential evapotranspiration ($E_0$). As explained in L. 151-153, the quantile regression model was fitted to binned $E, D$ data and we only used $E, D$ bins with a minimum of 100 observations (i.e. grid cells). This means that the estimate of $F_{0.99}$ for a particular $E, D$ bin was based on a sample of at least 100 20-year observation periods, which should provide a robust estimate of $F_{0.99}$, even for climates characterised by long fire return intervals, as long as fire events are a stationary process.

When looking globally at burnt area, and especially at extremes, one tends to only see Africa, which has so much burnt area than any other continent that it tends to completely dominate any analysis. In your methods you implement a bootstrapping, but I do wonder how different your F99 estimates would look without Africa (or the other direction, how much does it matter to include the rest of the world in the analysis?). This is always a problem, and no criticism, but for interpretation of the results it could be nice to know this kind of "uncertainty".

**Response:** As explained in section 2.3.1. (L. 147-155) our bootstrapping approach was designed to ensure that we sampled evenly across hydroclimatic space, thus avoiding dominance of observations from tropical savannas in Africa and other continents where much of global fire occurs. We have added some clarification on this point to the Discussion (L. 191-192): "Our model fitting approach, which sought to avoid dominance of the most common fire-prone environments on Earth such as tropical savannas, will have contributed to the relatively good performance of the model across a broad range of hydroclimatic conditions."

Minor comments
L8: maybe add human, as population density is supposed to be a good indicator of ignitions and suppression.
**Response:** L8 has been adapted, now referring to human activity.

L12-13: 99 percentile over? Time/space? It should be explicitly indicated how this is calculated. After reading the methods section and your previous manuscript over Australia I notice that it is a 99 percentile quantile regression, you should make this clearer in the sections which come before the methods (and possibly even for the results for people who don't want to go over the methods to interpret the results).
**Response:** The introduction provides a detailed definition of $F_{0.99}$ (see L. 62-75)

L79-80: P and Eo are not yet defined.
**Response:** $P$ and $E_0$. are defined in L.66-67.

L111-113: Does it matter that P and Eo come from different sources, e.g. a physical disconnection between both could influence your estimates for D? L212-214: I think this separation between production and dryness is nice. However, shouldn't there be a precipitation level where the default it is fuel limited? E.g. NPP is very low and almost exclusively limited by precipitation under very dry conditions. Now it seems that under even very low precipitation values it can you still be moisture limited?
**Response:** At very low mean annual $P$ and $E_0$ we observe very low levels of fire activity and the fitted response surface (Fig. 1C) is therefore relatively flat which implies that the position of the boundary between the two domains is relatively uncertain.
We do not expect areas of (very) low $P$ to fall by default in the domain of PL-type fire, because that depends on the level of $E_0$. Even at low mean annual $P$, sites can be humid if mean annual potential evapotranspiration is very low too due to cold temperature, low radiation inputs, etc.

L275: For a quantification of the spatial uncertainty in fire models you might be interested in this recent paper: https://www.geosci-model-dev-discuss.net/gmd-2019-261/
**Response:** Thanks, noted.

L290-294: I do follow the logic in using D for multiannual mean fire conditions (and I agree that the Nesterov Index is suboptimal), I don't see the logic in comparing predictions which are generally made at (sub)daily timesteps to your multiannual average F(99) estimates. These seem to be two pretty disconnected things and I don't see how you could use D for these short timestep responses in burnt area.
**Response:** In the Discussion section (L. 296-323) we are pointing out that the Nesterov Index is a poor predictor of fuel moisture content and suggesting that this could be a reason for poor performance by some of the existing global fire models in forest biomes where fire is primarily constrained by the frequency and duration of dry fuel periods. We did not suggest that the mean annual climatic water deficit would be an option to model daily fuel moisture conditions in global fire models. However, we do show in Supplementary Material S4 that mean annual climatic water deficit is strongly and linearly related to the mean annual frequency of days of predicted dead fuel moisture (DFMC) below 10%. We have added a statement (L. 307-310) that the DFMC model we used for that analysis in Supplementary Material S4 (Resco De Dios et al. 2015, doi.org/10.1016/j.agrformet.2015.01.002) would be an alternative to the Nesterev Index for modelling daily fuel moisture content in global fire models.

L310: Another comment about regarding this dichotomy between fuel and dryness, there was a recent paper by Alvarado et al which investigated this across the tropics and found important differences between continents, but only looking at precipitation. I wondered whether these results could explain these differences by looking to over an aridity gradient?
**Response:** Alvarado et al. (2019, 10.1111/geb.13034) based their classification of fire regimes in tropical savannas and grasslands being fuel- or moisture-limited entirely on whether sites had a positive or negative relationship between interannual variation in burned area and precipitation. Their approach did not lead to a clear dichotomy or mapping of fuel- and moisture limited fire regime as areas of positive and negative relationships with precipitation were found to occur within the same biome. We agree with the reviewer that the underlying reason for Alvarado et al.'s lack of separation between fuel- and moisture-limited fire regimes is likely because they did not take variation in potential evapotranspiration in to account. As shown in our Figure 1D, potential mean annual burned area ($F_{0.99}$) can either increase or

decrease with mean annual precipitation depending on the level of mean annual potential evapotranspiration. This shows that the dichotomy between fuel production and fuel dryness limitations is a function of the combination of climatic availability of water and energy.